# Factors associated with malnutrition inflammation score among hemodialysis patients: A cross-sectional investigation in tertiary care hospital, Palestine

Zakaria Hamdan[1]\*, Zaher Nazzal[1], Fatima Masoud Al-Amouri[2], Sanaa Ishtayah[1], Sarah Sammoudi[1], Lawra Bsharat[1], Manal Badrasawi[2]\*

1 Faculty of Medicine and Health Sciences, Department of Medicine, An-Najah National University, Nablus, Palestine, 2 Department of Nutrition and Food Technology, An-Najah National University, Nablus, Palestine

\* m.badrasawi@najah.edu (MB); z.hamdan@najah.edu (ZH)

**Data Availability Statement:** data are available on figshare repository, https://figshare.com/s/99bd7813d98ac1d114ca.

## Abstract

Malnutrition is a prevalent complication in hemodialysis patients and is associated with increased mortality and morbidity. This study aimed to identify the risk factors associated with malnutrition among hemodialysis patients including patient's general characteristics, functional status, and dietary intake. This study involved hemodialysis patients in An-Najah National University Hospital at Nablus/Palestine. An interview-based questionnaire was used to collect data related to sociodemographic, lifestyle, hemodialysis, medical history, anthropometrics, biochemical indices, dietary data using 3-days diet recall, and functional status. Malnutrition-inflammation score tool (MIS) was used for malnutrition screening of the studied patients. The study involved 188 patients, with a mean age of 57.8±14 years. A total of 28.2% participants are reported malnourished. Malnutrition was significantly associated with being female (p = 0.001), unemployed (p = 0.009), nonsmoker or ex-smoker (p = 0.018). Patients with CVDs (p = 0.006), higher months on dialysis (p = 0.002), lower BMI (p = 0.018), and using catheter for dialysis access are more likely to develop malnutrition (p = 0.018). Furthermore, patients with poor functional status (poor handgrip strength (p<0.001), inability to walk (p<0.001), needing help in daily activity or in transfer (p<0.001)) were significantly associated with malnutrition. Additionally, malnutrition was significantly higher among patients who does not meet their recommended calorie intake (p = 0.008), whose sodium intake within recommendation (p = 0.049), patients with low levels of hemoglobin (p = 0.022), albumin (p<0.001), TIBC (p = 0.002), phosphate (p<0.001), and patients with higher levels of ferritin (p<0.001). Higher months on dialysis (Exp(B) = 1.02), unmet calorie intake (Exp(B) = 4.3), needing help in daily activities (Exp(B) = 0.238), and low albumin level (Exp(B) = 0.048) were the independent predictors of malnutrition. This study highlights the significant association between malnutrition and various demographic, clinical, functional, and dietary factors in hemodialysis patients, revealing the burden of malnutrition during HD. It also necessitates targeted intervention to address these risk factors and improve nutritional status and the overall health outcomes of HD patients.

**Funding:** The author(s) received no specific funding for this work.

**Competing interests:** The authors have declared that no competing interests exist.

## Introduction

End-stage renal disease (ESRD) is an important health issue caused by aging populations and the widespread of chronic noncommunicable diseases [1]. ESRD results in decreased life expectancy and poor quality of life (QoL), leading to a tremendous financial strain on families and society in general [2,3]. The evolution of hemodialysis (HD) as one of the maintenance therapies for ESRD increases the survival rates of ERDS patients and improves their QoL significantly [4].

Despite the fact that HD is the major global treatment method for ERDS patients, representing about 89% worldwide, these patients are at high risk of morbidity and mortality due to malnutrition [5,6], which is considered a prevalent and life-threatening health concern in HD patients [7]. As a result, it is recommended to acquire a sufficient intake of energy and protein to avoid malnutrition among this population [8]. Moreover, inadequate dietary intake is considered the most important determinant and key factor in malnutrition during HD [9]. However, clinical practice guidelines recommend restricting potassium, sodium, and phosphate intake to avoid elevated electrolyte levels and their associated complications [8].

In Palestine, a previous study showed that 47.1% of the HD patients had mild-to-moderate malnutrition [10]. Malnutrition in HD patients develop through various mechanisms, including nutrient loss caused by dialysis, inflammation, multiple dialyzer reuse, metabolic acidosis, and dialysis adequacy, which are inescapable iatrogenic factors that significantly contribute to malnutrition. On the other hand, non-iatrogenic factors include inadequate dietary intakes, taste changes, poor appetite, insulin resistance, and psychosocial factors [11].

Nutritional status assessment and identification of nutritional indicators related to morbidity and mortality would allow for early and timely clinical intervention among HD patients [12]. Furthermore, maintaining a good functional status in patients with ESRD is an urgent need for physicians since it represents a significant predictor of the patient's QoL [13]. Functional status is defined as the ability to perform basic daily living activities and is associated with adverse outcomes when impaired, such as hospitalization, dependence on caregivers, poor QoL, and eventually death [13,14].

Few studies were conducted in Palestine among HD patients to determine the prevalence of malnutrition and its associated factors [10,15,16]. Our study is the first study in Palestine to focus on dietary assessment and intake. Additionally, it is the first study in Palestine to use the MIS tool.

The goal of this study was to comprehensively report functional status, dietary intake, and their association with malnutrition among HD patients. This study is expected to help guide the medical team in dealing with this group of patients by introducing interventional methods to achieve optimal nutritional status in HD patients.

## Materials and methods

### Study design and population

The present study is a hospital-based cross-sectional study conducted in hemodialysis unit of An-Najah National University Hospital (NNUH) in Nablus city. NNUH, a teaching tertiary referral hospital, houses the largest dialysis unit in the West Bank of Palestine. The sample size was estimated using the Raosoft sample calculator with a margin of error of 5% and a confidence level of 95%. A total of 196 patients were asked to participate voluntarily in the study; all of which agreed to participate and gave their informed consent. Eight patients were excluded due to missing data, and therefore, a total of 188 patients were involved in this study.

The inclusion criteria were patients who were aged over 18 years old and had been on maintenance HD for at least three months. Exclusion criteria were patients who were

anticipated to receive kidney transplant within six months or had a medical condition like cancer that could have an impact on their nutritional status.

### Data collection and research tools

Data was collected from November 1st, 2021, to January 31st, 2022, through face-to-face interviews. For each patient, interviews were conducted, and their medical records accessed and reviewed to collect data relevant to the study, as follows:

### Sociodemographic data

Sociodemographic data included age, gender, marital status, residency, level of education, smoking status, employment status, and family income.

### Clinical data

Clinical data included comorbidities (hypertension, diabetes, and CVDs), and hemodialysis-related data (dialysis duration in months, dialysis duration in minutes, and dialysis access).

### Biochemical indices

Biochemical indices included albumin, calcium, carbon dioxide ($CO_2$), hemoglobin (Hgb) level, ferritin, transferrin, chloride, glucose, total iron binding capacity (TIBC), phosphate, potassium, sodium, and parathyroid hormone (PTH), which recruited from patient's records.

### Malnutrition assessment

Malnutrition-Inflammation Score (MIS) was used for malnutrition assessment in this study. MIS is a comprehensive scoring system with significant association with hospitalization and mortality. It measures nutrition state, inflammation, and anemia in HD patients. The MIS tool is an accurate and reliable structured screening tool for the evaluation of malnutrition [5,6,12]. It is a good predictor of dialysis outcome and an indicator of malnutrition among HD patients [12]. The MIS tool consists of four main sections: a patient's medical history, physical examination, body mass index (BMI), and laboratory values. The four main sections are subdivided into 10 components, scored from 0 (normal) to 3 (very severe), with a total score ranges from 0 to 30 [12,17]. The cutoff point was 7, any score >7 considered malnutrition and any score ≤7 considered well-nourished [18].

### Functional status assessment

The study questionnaire includes three self-reported questions (related to the ability to walk, need help in daily activity, and need help in transfer) designed to assess poor functional status, which is defined by any of the three co-morbid conditions as specified by the Centers for Medicare and Medicaid Services (CMS) in Form CMS-2728 –inability to ambulate, inability to transfer or need of assistance with daily activities [19]. In addition, handgrip strength test was used to assess participants physical functions, which measured using the digital hand dynamometer. The results were obtained by taking the average of three readings for each patient [20].

### Diet assessment

Three days' diet recall was used for diet assessment. Data collected through personal interviews and phone call, in addition to food diary for three days: dialysis day, non-dialysis day, and a weekend day. The data was taken from the patients or their caregivers if the patient have any

abnormalities, such as deafness or cognitive impairment. Dietary intake data were collected and analyzed using the Palestinian food composition database (Pal nut) after permission is approved from the owner [21]. The nutritional intake of study participants is considered inadequate when their energy intake is below 25 kcal/kg/day and their protein intake is lower than 1.0 g/kg/day [22].

## Statistical analysis

Data was analyzed using the statistical package for the social sciences (IBM-SPSS) statistical software version 21. Continuous variables were analyzed using descriptive statistics such as means and standard deviation, while categorical variables were described using percentages and frequencies. To investigate the relationship between continuous and categorical variables, the Chi-square or one-way ANOVA test were used where appropriate. The level of significance was set at p-value <0.05. Further analysis was performed using a binary logistic regression model, and the logistic regression assumptions were checked before conducting the analysis. The multicollinearity was checked using collinearity diagnostic tests. The Hosmer-Lemeshow goodness of fit test was employed to assess how well the model fits the data.

## Ethics

The Institutional Review Board (IRB) ethical committee of An-Najah National University approved the study protocol [Reference number: Med Sep 2021/78], while permissions and approval were obtained from An-Najah National University Hospital (NNUH) to interview HD patients and access their medical records, and all participants provided written informed consent. The extraction sheet for each patient was filled anonymously, data was coded and used only for research purposes. In addition, participants' privacy and confidentiality of the data we collected were ensured.

## Results

### Patients' recruitment

Fig 1 shows patients recruitment steps. Among a total of 314 patients receiving hemodialysis in the dialysis unit of NNUH, only 196 met the inclusion criteria, and 188 included in the final analysis.

### Nutritional status

This study revealed, using the MIS tool, that 53 (28.2%) (95% CI: 21.9–35.2%) of adult HD patients are malnourished and 135 (71.8%) are well nourished. The mean score for MIS was 5.95±3.15 and the median score was 5 (0–17).

### Sociodemographic, lifestyle, clinical characteristics, and their association with malnutrition

Table 1 describes sociodemographic and clinical characteristics of the included patients. The mean age of the HD patients was 57.81 years old, with more than half (62.2%) being males and 71.8% being married. Most patients had a high school education or less (81.9%), unemployed (80.3%), and lived in cities (45.2%). The mean BMI was 27.7±6.7 kg/m$^2$, and most of them stated that they are non-smoker (40.5%). Hypertension (82.4%), Diabetes (54.8%) and CVDs (46.3%) were the most frequent comorbidities. All patients in this study were dialyzed using high-flux (Fresenius) membranes. The average dialysis duration was 47±41.0 months; the majority of them had dialysis access via shunt or fistula (86.7%), and 179 (95.2%) dialyzed

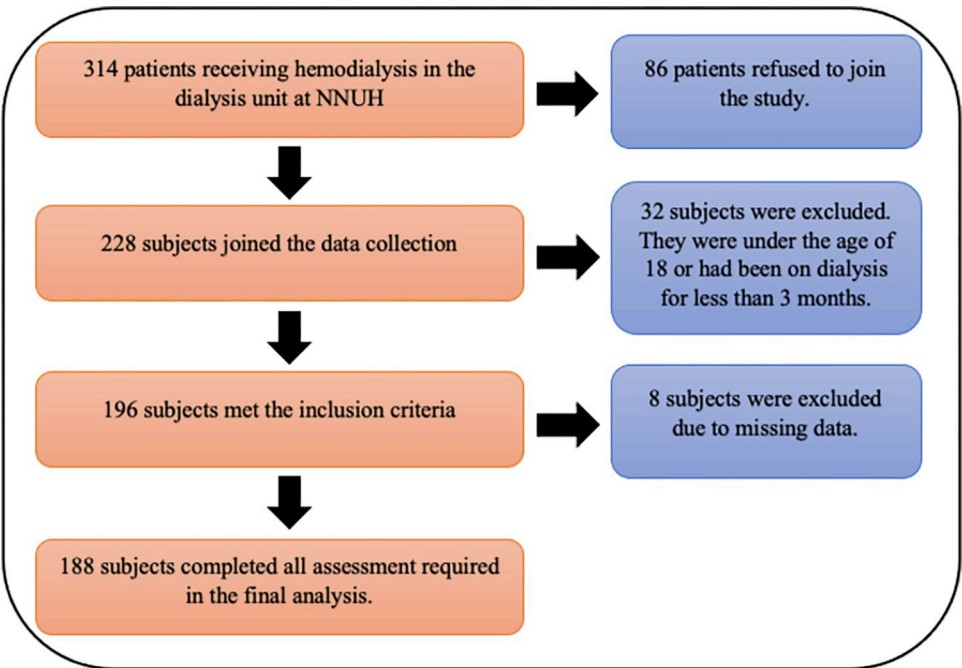

**Fig 1. Patient recruitment steps.**

thrice weekly. Univariate analysis revealed that gender (p = 0.001), employment status (p = 0.009), smoking status (p = 0.018), CVD (p = 0.006), BMI (p = 0.018), dialysis access (p = 0.018), and dialysis months (p = 0.002) were significantly associated with malnutrition, as presented in Table 1.

### Functional status, dietary intake, and their association with malnutrition

Table 2 details the functional status and dietary intake of HD patients. The ability to walk was only reported by 39.4% of the patients, while 36.7% and 26.1%, respectively, reported needing assistance with transfer and daily activities. Protein and calorie requirements were only met by 19.8% and 21.3% of the patients, respectively. On the other hand, Sodium intake exceeded recommendations among 54.8% of the patients. The comparison between patients with or without malnutrition on univariate analysis, revealed that malnutrition was significantly associated with patients who are not able to walk (p<0.001), need help in their daily actives or to transfer (p<0.001), and with their handgrip strength (p<0.001). In addition, malnutrition was significantly associated with calorie and sodium intake (p = 0.008 and 0.049, respectively), as presented in Table 2.

### Patients' biomedical data and its association with malnutrition

In this study, the mean value of hemoglobin was 11.6±1.2, serum albumin was 3.8±0.28, and 5.1±1.2 for phosphate level. In addition, serum albumin, hemoglobin, ferritin, TIBC, and phosphate levels were significantly associated with malnutrition, with a p-value of <0.001, 0.022, <0.001, 0.002, <0.001, respectively, as shown in Table 3.

### Binary logistic regression

The binary logistic model included all the significant variables found in the univariate analysis (gender, BMI, months of dialysis, employment status, smoking, CVDs, access of dialysis, handgrip

**Table 1. Patients' characteristics and malnutrition association.**

| Characteristics | | Total N = 188 | Malnutrition | | P-value |
| --- | --- | --- | --- | --- | --- |
| | | | Yes N = 53 (28.2%) | No N = 135 (71.8%) | |
| Age | | 57.8 ±14.0 | 60.8 ± 15.1 | 56.6 ± 13.5 | 0.065 |
| Body Mass Index | | 27.7 ± 6.7 | 25.9 ± 5.6 | 28.5 ± 6.9 | 0.018* |
| Ultrafiltration | | 3.0 ± 1.3 | 2.7 ± 1.2 | 3.1 ± 1.4 | 0.081 |
| Months on dialysis | | 47 ± 41.0 | 59.7 ± 40.4 | 42.1 ± 32.9 | 0.002* |
| Gender | Male | 117 (62.2%) | 23 (19.7%) | 94 (80.3%) | 0.001* |
| | Female | 71 (37.8%) | 30 (42.3%) | 41 (57.7%) | |
| Marital status | Married | 135 (71.8%) | 33 (24.4%) | 102 (75.6%) | 0.068 |
| | Unmarried | 53 (28.2%) | 20 (37.7%) | 33 (62.3%) | |
| Residency | City | 85 (45.2%) | 23 (27.1%) | 62 (72.9%) | 0.281 |
| | Village | 76 (40.4%) | 19 (25.0%) | 57 (75.0%) | |
| | Camp | 27 (14.4%) | 11 (70.7%) | 16 (59.3%) | |
| Educational level | High school or less | 154 (81.9%) | 47 (30.5%) | 107 (69.5%) | 0.133 |
| | Diploma or higher | 34 (18.1%) | 6 (17.6%) | 28 (71.8%) | |
| Employment status | Employed | 37 (19.7%) | 4 (10.8%) | 33 (89.2%) | 0.009* |
| | Unemployed | 151 (80.3%) | 49 (32.5%) | 102 (71.8%) | |
| Smoking status | Yes | 61 (32.4%) | 9 (14.8%) | 52 (85.2%) | 0.018* |
| | Ex-smoker | 51 (27.1%) | 18 (35.3%) | 33 (64.7%) | |
| | No | 76 (40.5%) | 26 (34.2%) | 50 (65.8%) | |
| Hypertension | Yes | 155 (82.4%) | 44 (28.4%) | 111 (71.6%) | 0.897 |
| | No | 33 (17.6%) | 9 (27.3%) | 24 (72.7%) | |
| Diabetes | Yes | 103 (54.8%) | 34 (33.0%) | 69 (67.0%) | 0.106 |
| | No | 85 (45.2%) | 19 (22.4%) | 66 (77.6%) | |
| CVDs | Yes | 87 (46.3%) | 33 (37.9%) | 54 (62.1%) | 0.006* |
| | No | 101 (53.7% | 20 (19.8%) | 81 (80.2%) | |
| Dialysis access | Catheter | 25 (13.3%) | 12 (48.0%) | 13 (52.0%) | 0.018* |
| | Shunt or fistula | 163 (86.7%) | 41 (25.2%) | 122 (74.8%) | |

Significant at

*: p<0.05, according to Chi square test.

strength, ability to walk, need help in daily activity, need help to transfer, calorie intake, sodium intake, albumin, hemoglobin, ferritin, TIBC, and phosphate levels). The Hosmer Lemeshow test for the final model showed that goodness of fit of the model was acceptable (p = 0.470); Cox & Snell R square was 0.428; and Nagelkerke R square was 0.614. According to this model, months of dialysis, needing help in daily activity, calorie requirement intake, and albumin level are the only predictors of malnutrition risk (p<0.05), i.e., increased months of dialysis increases the risk of having malnutrition (Exp(B) = 1.022, p<0.05, 95%CI = 1.007–1.037); needing help in daily activity lowers the risk of having malnutrition (Exp(B) = 0.238, p<0.05, 95%CI = 0.063–0.899); unmet calorie requirement intake increases the risk of having malnutrition (Exp(B) = 4.309, p<0.05, 95%CI = 1.075–17.263), and lower albumin level reduces the risk of developing malnutrition (Exp(B) = 0.048, p<0.05, 95%CI = 0.004–0.627), as shown in Table 4.

## Discussion

This study investigated malnutrition among hemodialysis patients using MIS and its relationship with their functional status, dietary intake, and general characteristics. The prevalence of

**Table 2. Dietary intake and functional status of hemodialysis patients in relation to malnutrition.**

| Characteristics | | Total N = 188 | Malnutrition | | P-value |
|---|---|---|---|---|---|
| | | | Yes N = 53 (28.2%) | No N = 135 (71.8%) | |
| **Handgrip strength** | | 17.8 ± 7.5 | 6.5 ± 11.5 | 22.4 ± 23.1 | <0.001* |
| **The ability to walk** | Yes | 74 (39.4%) | 10 (13.5%) | 64 (86.5%) | <0.001* |
| | No | 114 (60.6%) | 43 (37.7%) | 71 (62.3%) | |
| **Need help in daily activity** | Yes | 49 (26.1%) | 30 (61.2%) | 19 (38.8%) | <0.001* |
| | No | 139 (73.9%) | 23 (16.5%) | 116 (83.5%) | |
| **Need help in transfer** | Yes | 69 (36.7%) | 36 (52.2%) | 33 (47.8%) | <0.001* |
| | No | 119 (63.3%) | 17 (14.3%) | 102 (85.7%) | |
| **Calorie requirement** | Met (≥ 25 kcal/kg/day) | 40 (21.3%) | 18 (45.0%) | 22 (55.0%) | 0.008* |
| | Unmet (< 25 kcal/kg/day) | 148 (78.7%) | 35 (23.6%) | 113 (76.4%) | |
| **Protein requirement** | Met (≥ 1.0 g/kg/day) | 37 (19.8%) | 15 (40.5%) | 22 (59.5%) | 0.066 |
| | Unmet (< 1.0 g/kg/day) | 150 (80.2%) | 38 (25.3%) | 112 (74.7%) | |
| **Potassium intake** | Within recommendation | 178 (94.7%) | 51 (28.7%) | 127 (71.3%) | 0.554 |
| | Exceed recommendation | 10 (5.3%) | 2 (20.0%) | 8 (80.0%) | |
| **Sodium intake** | Within recommendation | 85 (45.2%) | 30 (35.3%) | 55 (64.7%) | 0.049* |
| | Exceed recommendation | 103 (54.8%) | 23 (22.3%) | 80 (77.7%) | |
| **Phosphorous intake** | Within recommendation | 162 (86.2%) | 48 (29.6%) | 114 (70.4%) | 0.274 |
| | Exceed recommendation | 26 (13.8%) | 5 (19.2%) | 21 (80.8%) | |
| **Calcium intake** | Within recommendation | 172 (91.5%) | 49 (28.5%) | 123 (71.5%) | 0.767 |
| | Exceed recommendation | 16 (8.5%) | 4 (25.0%) | 12 (75.0%) | |

Significant at

*: p<0.05, according to Chi square test.

**Table 3. Biochemical indices of hemodialysis patients in relation to malnutrition.**

| Biochemical indices[1] (Mean ± SD) | Total (SI unit) N = 188 | Malnutrition | | P-value |
|---|---|---|---|---|
| | | Yes N = 53 (28.2%) | No N = 135 (71.8%) | |
| **Albumin serum g/dl** | 3.8 ± 0.28 (38 g/l) | 3.6 ± 0.29 | 3.9 ± 0.23 | <0.001* |
| **Calcium mEq/l** | 8.8 ± 0.78 (4.4 mmol/l) | 8.7 ± 0.79 | 8.8 ± 0.75 | 0.666 |
| **Hemoglobin g/dl** | 11.6 ± 1.2 (116 g/l) | 11.2 ± 1.2 | 11.7 ± 1.2 | 0.022* |
| **Ferritin ng/ml** | 714.4 ± 372.3 (1605.26 pmol/l) | 896.8 ± 411.1 | 644 ± 331.5 | <0.001* |
| **Transferrin mg/dl** | 33.0 ± 10.8 (3.3 g/l) | 33.4 ± 12.1 | 32.9 ± 10.3 | 0.760 |
| **Chloride mEq/l** | 97.5 ± 3.2 (97.5 mmol) | 97.4 ± 3.2 | 97.5 ± 3.2 | 0.868 |
| **Glucose mg/dl** | 155.1 ± 69.0 (8.61 mmol/l) | 164 ± 73.4 | 151 ± 68.1 | 0.254 |
| **TIBC ug/dl** | 204.8 ± 33.1 (36.66 umol/l) | 192.7 ± 27.1 | 209.7 ± 34.2 | 0.002* |
| **Phosphate mg/dl** | 5.1 ± 1.2 (1.65 mmol/l) | 4.5 ± 1.1 | 5.3 ± 1.2 | <0.001* |
| **Potassium mEq/l** | 4.9 ± 0.59 (4.9 mmol/l) | 4.8 ± 0.66 | 4.9 ± 0.56 | 0.424 |
| **PTH pg/ml** | 445.2 ± 43.2 (445.2 ng/l) | 431.7 ± 529.7 | 450.6 ± 389.2 | 0.788 |
| **Sodium mEq/l** | 137.5 ± 2.6 (137.5 mmol/l) | 137.3 ± 2.7 | 137.7 ± 2.6 | 0.456 |

[1]Average readings of last 4 months. Significant at

*: p<0.05 according to one-way ANOVA.

**Table 4. Malnutrition risk factors using binary logistic regression analysis.**

| Factors (reference) | B | P-value | Exp (B)[a] | Confidence Interval | Exp (B)[a] | P-value of model |
|---|---|---|---|---|---|---|
| **Gender: females** (males) | 1.029 | 0.111 | 2.798 | 0.790–9.909 | 0.394 | <0.001** |
| **Low BMI** (continuous) | -0.072 | 0.121 | 0.931 | 0.851–1.019 | | |
| **Higher months of dialysis** (continuous) | 0.022 | 0.003* | 1.022 | 1.007–1.037 | | |
| **Unemployed** (employed and retired) | 0.470 | 0.602 | 1.600 | 0.274–9.352 | | |
| **Ex- and nonsmokers** (smokers) | -0.143 | 0.660 | 0.867 | 0.459–1.638 | | |
| **CVD** (no CVD) | -0.919 | 0.090 | 0.399 | 0.138–1.156 | | |
| **Dialysis access: catheter** (shunts or fistulas) | -1.273 | 0.124 | 0.280 | 0.055–1.417 | | |
| **Low handgrip strength** (continuous) | 0.011 | 0.625 | 1.011 | 0.969–1.054 | | |
| **Inability to walk** (able to walk) | -0.079 | 0.902 | 0.924 | 0.265–3.277 | | |
| **Need help in daily activity** | -1.435 | 0.034* | 0.238 | 0.063–0.899 | | |
| **Need help to transfer** | -0.491 | 0.482 | 0.612 | 0.156–2.402 | | |
| **Unmet calorie requirement** (met) | 1.461 | 0.039* | 4.309 | 1.075–17.263 | | |
| **Met sodium recommendation** (unmet) | -1.013 | 0.084 | 0.363 | 0.115–1.146 | | |
| **Low albumin level** (continuous) | -3.030 | 0.021* | 0.048 | 0.004–0.627 | | |
| **Low hemoglobin level** (continuous) | -0.327 | 0.102 | 0.721 | 0.487–1.067 | | |
| **High ferritin level** (continuous) | 0.001 | 0.249 | 1.001 | 0.999–1.003 | | |
| **Low TIBC level** (continuous) | -0.004 | 0.701 | 0.996 | 0.977–1.016 | | |
| **Low phosphate level** (continuous) | -0.440 | 0.108 | 0.644 | 0.376–1.102 | | |

*: $p<0.05$

**: $p<0.001$ using binary logistic regression.

[a]: Exponentiation of the B coefficient.

malnutrition diagnosed by MIS in this study was 28.2% among MHD patients. Previous studies that used MIS revealed that malnourished HD patients were up to 10.2% [23], 33% [24], 51% [25], 60% [26], and 71.7% [27]. This variation is attributed to different sample sizes and MIS cutoff points between studies.

A combination of factors is contributing to malnutrition, according to our findings. Gender was an important factor for malnutrition in HD patients in this study. Women are more affected by malnutrition at a significant level compared to men. A former study conducted in Iran reported a higher prevalence of malnutrition in women than men, indicating the necessity of nutritional status screening in women undergoing HD [28]. However, Aggarwal et al. and Badrasawi et al. found no relationship between gender and malnutrition in HD patient [16,26].

In this study, unemployment was another factor associated with malnutrition among HD patients. This finding was in agreement with previous findings [16,20,29], which are attributed to their low income, which might lead to limited dietary intake and poor treatment [20]. In addition, ex-smokers and non-smokers showed a higher prevalence of malnutrition in this study. In line with our findings, a previous study indicated that malnutrition was 2.26 times more prevalent among former smokers [30]. Conversely, Azzeh et al. found no difference between nourished and malnourished patients regarding their smoking status [20]. However, the high prevalence of malnutrition in non-smokers might be attributed to psychological effects or a potential bias in the study population.

According to the current study, BMI is one of the factors that has a significant relationship with malnutrition. Our analysis indicates a lower mean BMI among HD patients who were malnourished. This finding was in agreement with former research, which found that malnourished patients have a lower BMI in comparison to their well-nourished counterparts [31].

Meanwhile, the findings of this study indicated a significant association between CVD and malnutrition. According to previous findings, malnutrition is a risk factor for CVD in patients with CKD [32]. It is worth noting that 70–80% of MHD patients develop CVD, and CVD mortality is up to 10–30 times higher compared to the general population and is considered the most common complication among ESRD patients [33].

Our study revealed that malnutrition was significantly higher in patients who had a longer duration on hemodialysis compared to patients with a shorter duration. This result was supported by a former study conducted by Bramania et al. [31], where it was found that the longer duration of hemodialysis was associated with an increased risk of malnutrition.

In addition, our study found that patients who are using catheters for dialysis access represent a higher percentage of malnutrition compared to patients using shunts or fistulas. This could be explained by the fact that central catheters are especially associated with infection and septicemia [34], which are associated with a decreased QoL due to increased hospitalization [35]. All of which might contribute to disrupted eating patterns, inadequate nutrition, and dietary restrictions, leading to malnutrition among HD patients.

Regarding patients' functional status and malnutrition relationship, the present study found that malnutrition is significantly higher among patients with poor functional status, including those with lower handgrip strength, a lower ability to walk, and who need help in daily activity and transfer. Consistent with our findings, a former study revealed a significant association between lower handgrip values and malnutrition, which indicated the nutritional status deterioration and the potential muscle weakness attributed to the impact of malnutrition among HD patients [20]. Another study found that the high risk of malnutrition among HD patients was significantly higher in patients who were not able to ambulate. However, the same study found no significant association between patients who need help in their daily activity or to transfer and malnutrition [16]. In addition, an earlier study confirmed the usefulness of malnutrition scores in predicting functional impairment in HD patients [36].

In the current study, a significant difference in malnutrition was found between patients who met their calorie requirements intake and patients who did not meet their calorie requirements intake, while calorie intake less than recommendation increases the risk of developing malnutrition by 4.3 folds, according to our findings. A previous study found no difference in dietary calorie intake between well-nourished and malnourished patients [37]. On the other hand, maintaining adequate protein and energy intake is an important factor in avoiding malnutrition in dialysis patients [11]. Additionally, a significant association was found between the prevalence of protein energy wasting and the dietary intake of energy, whereas energy and protein intake lower than the recommended results in malnutrition among HD patients [38]. Among other dietary factors, the present study found a significant association between malnutrition and patients whose sodium intake is within the recommended range compared to patients whose sodium intake exceeds the recommended range. Previous findings indicated that low sodium levels, known as hyponatremia, are frequently associated with malnutrition in chronic HD patients [39]. Our findings might be attributed to the imbalance in dietary intake among study participants, in which they may follow the sodium recommendation by focusing on reducing sodium at the expense of their overall nutrient' intake, which necessitates dietary education and support to ensure dietary adequacy among malnourished HD patients.

Patients' biochemical data was found to have a significant difference between nourished and malnourished individuals in this study, including albumin, hemoglobin, TIBC, ferritin, and phosphate levels. In line with our findings, previous studies showed an inversely significant relationship between albumin levels and malnutrition in HD patients [28]. Moreover, other studies found that a low albumin level in hemodialysis patients is a strong predictor that reflects malnutrition [5,40]. Even though it was thought that a low serum albumin level was

related to inflammation rather than malnutrition among HD patients [40]. However, malnourished patients in this study showed lower hemoglobin and TIBC levels and a higher level of ferritin at a significant level. Unlike our findings, previous studies found no significant relationship between malnutrition and hemoglobin levels [16,20,31,41], which might be attributed to different sample sizes and the sensitivity of malnutrition assessment tools between studies. However, it is well documented that low hemoglobin levels are a common health implication among patients on dialysis [42]. On the other hand, in line with our results, it was found that increased ferritin levels and low TIBC levels are independent factors associated with all-cause mortality in ESRD patients, and low TIBC levels are associated with iron deficiency and protein-energy wasting in MHD patients, while high ferritin increases inflammation [43]. Interestingly, serum ferritin is widely recognized as an acute-phase reactant, frequently elevated in CKD patients as a result of systematic inflammation, and correlates positively with the severity of inflammation [44,45]. In addition, lower phosphate levels were significantly associated with malnutrition among HD patients in this study, which is in line with a former study that found that hypophosphatemia (<3.5 mg/dl) is a marker associated with malnutrition in patients undergoing hemodialysis [46]. Moreover, another former study found that a low phosphate level may be a sign of impending malnutrition [47].

It's worth noting that our multivariate analysis, unlike the univariate analysis, revealed a protective effect of being in need of help in daily activity and having a low albumin level against developing malnutrition (Exp (B) <1, 95%CI <1), which might seem counterintuitive. However, it is possible that patients with these conditions receive more nutritional care and support, leading to better nutritional status and lowering their risk of being malnourished. Otherwise, more comprehensive research is required to understand this intriguing observation. Nevertheless, further explanations for our findings regarding the protective effect of low albumin levels against malnutrition may include confounding factors, which might influence the association and mask the link between low albumin and malnutrition. In addition, this finding could be explained by selection bias, in which healthier patients with mild hypoalbuminemia might be included in this study.

This study has some limitations, including limited generalizability of findings because of the single-center study setting, the nature of cross-sectional studies that limit identifying cause-effect relationships, the self-reported data making it prone to recall bias, the potential selection bias, and the exclusion of certain patients' groups. Moreover, insufficient data on C-reactive protein (CRP), type of dialysis membrane, and dialysis adequacy limit the capacity to assess their potential link to nutritional status among HD patients in this study. Including such information could provide significant insights and should be considered for future investigations. Regardless of these limitations, our research presents comprehensive data collection, uses validated assessment tools, and uses detailed diet assessment, which is crucial to understanding nutritional status.

## Conclusions

This study provides a thorough understanding of the determinants contributing to malnutrition among HD patients. It highlights the significant link between malnutrition and demographic, clinical, functional, and nutritional factors, revealing the serious burden of malnutrition during HD. Malnutrition was more prevalent among females, the unemployed, and non-smokers or ex-smokers. Patients with cardiovascular diseases, a lower BMI, catheter access for dialysis, and those who had been on dialysis for a longer period were more likely to develop malnutrition. In addition, a strong link was found between malnutrition and functional impairments, including poor handgrip strength, an inability to walk, and the need for

help in daily activities or transfers, which reflect the influence of nutritional status on physical activity. Moreover, lower albumin, hemoglobin, phosphate, TIBC, and elevated ferritin suggest a biochemical imbalance accompanying malnutrition, potentially deteriorating their overall health state. Moreover, dietary factors play a crucial role in malnutrition. Patients who did not consume their recommended calorie intake but had a sodium intake within the recommended range were more likely to be malnourished. Regression analysis showed that unmet calorie intake increased the risk of malnutrition, which necessitates nutritional support and education in HD patients. Therefore, it is critical to provide a targeted intervention for HD patients, including dietary improvement and clinical and functional impairment management, which are essential to mitigate malnutrition and improve the outcomes of HD patients. Further research is recommended to explore the cause-effect relationship for proper future intervention.

## Acknowledgments

We express our gratitude to every patient participate in this study.

## Author Contributions

**Data curation:** Zakaria Hamdan, Zaher Nazzal, Sanaa Ishtayah, Sarah Sammoudi, Lawra Bsharat.

**Formal analysis:** Zakaria Hamdan, Zaher Nazzal, Fatima Masoud Al-Amouri, Sanaa Ishtayah, Sarah Sammoudi, Lawra Bsharat.

**Investigation:** Sanaa Ishtayah, Sarah Sammoudi, Lawra Bsharat.

**Methodology:** Zakaria Hamdan, Zaher Nazzal.

**Project administration:** Sanaa Ishtayah, Sarah Sammoudi, Lawra Bsharat, Manal Badrasawi.

**Supervision:** Manal Badrasawi.

**Writing – original draft:** Fatima Masoud Al-Amouri.

**Writing – review & editing:** Manal Badrasawi.

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
