## [Decision Letter · Decision Letter 0]

21 Nov 2024

PONE-D-24-30690Factors Associated with Malnutrition Inflammation Score Among Hemodialysis Patients: A Cross-Sectional Investigation in Tertiary Care Hospital, PalestinePLOS ONE

Dear Dr. Badrasawi,

Thank you for submitting your manuscript to PLOS ONE. After careful consideration, we feel that it has merit but does not fully meet PLOS ONE’s publication criteria as it currently stands. Therefore, we invite you to submit a revised version of the manuscript that addresses the points raised during the review process.

We look forward to receiving your revised manuscript.

Kind regards,

Ahmet Murt

Academic Editor

PLOS ONE

Journal Requirements:

Additional Editor Comments:

In this study, authors analyzed factors that were related to malnutrition - inflammation score among hemodialysis patients. They involved 188 patients 28,2% of whom were malnourished. Our impartial reviewers raised some major points to be revised in this manuscript. Please provide these additional data.

Reviewers' comments:

Reviewer's Responses to Questions

**Comments to the Author**

1. Is the manuscript technically sound, and do the data support the conclusions?

Reviewer #1: Yes

Reviewer #2: Yes

2. Has the statistical analysis been performed appropriately and rigorously? 

Reviewer #1: Yes

Reviewer #2: I Don't Know

3. Have the authors made all data underlying the findings in their manuscript fully available?

Reviewer #1: Yes

Reviewer #2: Yes

4. Is the manuscript presented in an intelligible fashion and written in standard English?

Reviewer #1: Yes

Reviewer #2: Yes

5. Review Comments to the Author

Reviewer #1: The paper titled " Factors Associated with Malnutrition Inflammation Score Among Hemodialysis Patients: A Cross-Sectional Investigation in Tertiary Care Hospital, Palestine" need minor revision of the affiliation section, as I think it was repeated. The scientific content is good and it highlight a major cause of death in hemodialysis patients.

Reviewer #2: 1. Did all patients included in the study use the same type of membrane for hemodialysis? Were there any differences, such as the use of high-flux, low-flux, or MCO membranes? Is there any data on this? If so, could you please share it?

2. The removal of uremic solutes depends on dialysis adequacy. Expert guidelines for optimal uremic solute removal recommend a dialysis adequacy target of achieving a Kt/Vurea of 1.2. Due to dialysis inadequacy, conditions such as uremic toxins and acidosis affect both protein catabolism and appetite. Is there any data on hemodialysis adequacy between the malnutrition and non-malnutrition groups? What are the Kt/V values in these two groups?

3. Of the 188 patients included in the study, 113 undergo hemodialysis twice a week. We know that as hemodialysis duration and frequency increase, better solute clearance is achieved. In the malnutrition group, were most patients undergoing hemodialysis twice a week? Was there a difference in the frequency of hemodialysis sessions between the groups?

4. Is there any data on CRP, an inflammatory marker? Is there a difference in CRP levels between the malnutrition group and the non-malnutrition group? If so, could you please share it?

5. For HD patients, dietary energy intake (DEI) and dietary protein intake (DPI) recommendations for adults have been proposed by various expert groups. These generally range from 25–35 kcal/kg ideal body weight (IBW)/day for DEI and 1.0–1.2 g protein/kg IBW/day for DPI. What is your definition of inadequate nutrient intake?

6. The dialysis process leads to chronic nutrient losses, particularly of proteins and amino acids. Hypoalbuminemia is a strong predictor of malnutrition. According to your study, a lower albumin level was found to reduce the risk of developing malnutrition. How can you explain this?

6. PLOS authors have the option to publish the peer review history of their article (what does this mean?). If published, this will include your full peer review and any attached files.

Reviewer #1: **Yes: **AHMED AKL

Reviewer #2: No

---

## [Author Response · Author response to Decision Letter 0]

4 Dec 2024

Dear editors and reviewers, 

We kindly thank you for your time, effort, and time invested during the review of our manuscript. We appreciate the comments and suggestions on the submitted version of our contribution. On the basis of those, we have produced a revised and improved version, which we are submitting to PLOS ONE for an additional review. 

We have formulated detailed answers to the reviewers comments, which is uploaded in point-to-point replies in the attached files.

---

## [Decision Letter · Decision Letter 1]

22 Dec 2024

Factors Associated with Malnutrition Inflammation Score Among Hemodialysis Patients: A Cross-Sectional Investigation in Tertiary Care Hospital, Palestine

PONE-D-24-30690R1

Dear Dr. Badrasawi,

We’re pleased to inform you that your manuscript has been judged scientifically suitable for publication and will be formally accepted for publication once it meets all outstanding technical requirements.

Kind regards,

Ahmet Murt

Academic Editor

PLOS ONE

Additional Editor Comments (optional):

In this study that analyzed the factors that are associated with Malnutrition Inflammation Score in maintenance hemodialysis patients, the authors well replied to all reviewer comments.

Reviewers' comments:

Reviewer's Responses to Questions

**Comments to the Author**

1. If the authors have adequately addressed your comments raised in a previous round of review and you feel that this manuscript is now acceptable for publication, you may indicate that here to bypass the “Comments to the Author” section, enter your conflict of interest statement in the “Confidential to Editor” section, and submit your "Accept" recommendation.

Reviewer #2: All comments have been addressed

Reviewer #3: All comments have been addressed

2. Is the manuscript technically sound, and do the data support the conclusions?

Reviewer #2: Partly

Reviewer #3: Yes

3. Has the statistical analysis been performed appropriately and rigorously? 

Reviewer #2: Yes

Reviewer #3: Yes

4. Have the authors made all data underlying the findings in their manuscript fully available?

Reviewer #2: Yes

Reviewer #3: Yes

5. Is the manuscript presented in an intelligible fashion and written in standard English?

Reviewer #2: Yes

Reviewer #3: Yes

6. Review Comments to the Author

Reviewer #2: First of all, I would like to thank the authors for their responses. I think the article has improved with these corrections.

Reviewer #3: The authors replied to all reviewers comments from the previous round of review. I have no further comments.

7. PLOS authors have the option to publish the peer review history of their article (what does this mean?). If published, this will include your full peer review and any attached files.

Reviewer #2: No

Reviewer #3: No

---

## [Editor Report · Acceptance letter]

15 Jan 2025

PONE-D-24-30690R1 

PLOS ONE

Dear Dr. Badrasawi, 

I'm pleased to inform you that your manuscript has been deemed suitable for publication in PLOS ONE. Congratulations! Your manuscript is now being handed over to our production team.

Kind regards, 

on behalf of

Dr. Ahmet Murt 

Academic Editor

PLOS ONE